# Fusion rules and structure constants of E-series minimal models

**Rongvoram Nivesvivat[1] and Sylvain Ribault[2]**

[1] *New York University Abu Dhabi, United Arab Emirates*
[2] *Institut de physique théorique, CEA, CNRS, Université Paris-Saclay*

*E-mail:* `rongvoram.n@outlook.com`, `sylvain.ribault@ipht.fr`

ABSTRACT: In the ADE classification of Virasoro minimal models, the E-series is the sparsest: their central charges $c = 1 - 6\frac{(p-q)^2}{pq}$ are not dense in the half-line $c \in (-\infty, 1)$, due to $q = 12, 18, 30$ taking only 3 values — the Coxeter numbers of $E_6, E_7, E_8$. The E-series is also the least well understood, with few known results beyond the spectrum.

Here, we use a semi-analytic bootstrap approach for numerically computing 4-point correlation functions. We deduce non-chiral fusion rules, i.e. which 3-point structure constants vanish. These vanishings can be explained by constraints from null vectors, interchiral symmetry, simple currents, extended symmetries, permutations, and parity, except in one case for $q = 30$. We conjecture that structure constants are given by a universal expression built from the double Gamma function, times polynomial functions of $\cos(\pi\frac{p}{q})$ with values in $\mathbb{Q}\left(\cos(\frac{\pi}{q})\right)$, which we work out explicitly for $q = 12$.

We speculate on generalizing E-series minimal models to generic integer values of $q$, and recovering loop CFTs as $p, q \to \infty$.

# 1 Introduction

## 1.1 How exceptional are E-series minimal models?

Virasoro minimal models are the two-dimensional conformal field theories that obey the following assumptions:

- **Rationality**: The spectrum is made of finitely many indecomposable representations of the conformal algebra.

- **Irreducibility** of all representations that appear.

- **Modular invariance** of the torus partition function.

The CFTs that obey these assumptions can be classified, leading to the ADE classfication of minimal models [1]. Minimal models are parametrized by coprime integers $p, q \in \mathbb{N} + 2$ that determine their central charges

$$c_{p,q} = 1 - 6\frac{(p-q)^2}{pq} \ . \tag{1.1}$$

Moreover, these integers obey the following conditions:

| Model | Condition | #{irreducible representations} |
|-------|-----------|-------------------------------|
| $A_{p,q}$ | $q > p$ | $\frac{1}{2}(p-1)(q-1)$ |
| $D_{p,q}$ | $q \in 4\mathbb{N}+6$ | $\frac{1}{2}(p-1)q$ |
|  | $q \in 4\mathbb{N}+8$ | $\frac{1}{2}(p-1)(q-1)$ |
| $E_{p,q}$ | $q = 12$ | $\frac{1}{2}(p-1)q$ |
|  | $q = 18$ | $\frac{1}{2}(p-1)(q-1)$ |
|  | $q = 30$ | $\frac{1}{2}(p-1)(q+2)$ |

(1.2)

A-series and D-series minimal models have a prominent place in the landscape of solvable CFTs [2], thanks to the following features:

- The spectrums of A-series and D-series minimal models emerge naturally from the representation theory of the Virasoro algebra at rational central charge.

- Their 3-point structure constants are known explicitly, so they can be considered solved.

- The central charges of the A-series and D-series minimal models are dense in the half-line $c \in (-\infty, 1)$. This allows us to take non-rational limits in $c$, and find consistent non-rational CFTs. The converse operation of recovering these minimal models as limits from generic $c$ is also possible to some extent.

- Some A-series and D-series minimal models describe critical limits of lattice models that are simple and interesting, such as the Lee–Yang, Ising and 3-state Potts models.

The situation is less favourable for E-series minimal models: their spectrums are complicated and strongly $q$-dependent, their structure constants are not known explicitly, their central charges are not dense, and they do not have compelling applications, although some of them describe critical limits of integrable lattice models [3], [4, Appendix E].

We could consider E-series minimal models as eccentric oddities. However, the dearth of exactly solvable CFTs motivates us to entertain another possibility: **By relaxing rationality, irreducibility and/or modular invariance, could we extend the E-series to infinitely many values of $q$?**

To extend the E-series, natural candidates would be CFTs whose primary fields belong to the extended Kac table, i.e. their Kac indices would be integer but unbounded. We would expect such CFTs to be irrational and logarithmic. They have sometimes been called logarithmic minimal models, although that expression has never been defined clearly [5], and has also been applied to loop CFTs [2] at $c = c_{p,q}$, which have fractional Kac indices.

Before being able to conclusively answer our question, we however need to better understand the E-series minimal models themselves. In this article we will use bootstrap techniques from loop CFTs, in order to determine fusion rules and structure constants. In the process we will find similarities between loop CFTs and E-series minimal models at the levels of spectrums and structure constants.

## 1.2 The spectrums' tale

At a given central charge $c_{p,q}$, there may exist up to 3 different minimal models belonging to different series. Consider irreducible, fully degenerate representations of the Virasoro algebra $\mathcal{R}_{r,s}$, parametrized by Kac indices $r, s$, and characterized by their conformal dimensions

$$\Delta_{r,s} = \frac{(pr - qs)^2 - (p-q)^2}{4pq} \ . \tag{1.3}$$

The spectrums of minimal models are built from a set of such representation that is closed under fusion: $\{\mathcal{R}_{r,s}\}_{(r,s)\in\frac{1}{2}]0,q[\times]0,p[}$, where the factor $\frac{1}{2}$ accounts for the invariance of $\mathcal{R}_{r,s}$ under $(r,s) \mapsto (q-r, p-s)$, and we introduce the notation

$$b - a \in \mathbb{N} \quad \implies \quad ]a, b[ = \{a+1, a+2, \ldots, b-1\} \ . \tag{1.4}$$

The space of states of $A_{p,q}$ is simply the diagonal combination $\frac{1}{2} \bigoplus_{r\in]0,q[} \bigoplus_{s\in]0,p[} |\mathcal{R}_{r,s}|^2$, where $|\mathcal{R}|^2 = \mathcal{R} \otimes \bar{\mathcal{R}}$ is the tensor product of the representation $\mathcal{R}$ of the left-moving Virasoro algebra, with the same representation of the right-moving Virasoro algebra. The space of states of $D_{p,q}$ is not much more complicated, as $D_{p,q}$ can be interpreted as a $\mathbb{Z}_2$ orbifold of $A_{p,q}$. However, the space of states of $E_{p,q}$ is more mysterious, and strongly depends on $q$:

$$\mathcal{S}_{p,12} = \bigoplus_{\substack{s\in]0,p[\\s\in2\mathbb{Z}+1}} |\mathcal{R}_{1,s} \oplus \mathcal{R}_{7,s}|^2 \oplus \frac{1}{2} \bigoplus_{s\in]0,p[} |\mathcal{R}_{4,s} \oplus \mathcal{R}_{8,s}|^2 \oplus \bigoplus_{\substack{s\in]0,p[\\s\in2\mathbb{Z}}} |\mathcal{R}_{5,s} \oplus \mathcal{R}_{11,s}|^2 \ , \tag{1.5a}$$

$$\mathcal{S}_{p,18} = \frac{1}{2} \bigoplus_{s\in]0,p[} \left\{ |\mathcal{R}_{9,s} \oplus 2\mathcal{R}_{3,s}|^2 \ominus 4|\mathcal{R}_{3,s}|^2 \oplus \bigoplus_{r\in\{1,5,7\}} |\mathcal{R}_{r,s} \oplus \mathcal{R}_{18-r,s}|^2 \right\} \ , \tag{1.5b}$$

$$\mathcal{S}_{p,30} = \frac{1}{2} \bigoplus_{s\in]0,p[} \left\{ \left| \bigoplus_{r\in\{1,11,19,29\}} \mathcal{R}_{r,s} \right|^2 \oplus \left| \bigoplus_{r\in\{7,13,17,23\}} \mathcal{R}_{r,s} \right|^2 \right\} \ . \tag{1.5c}$$

In the cases $q = 12$ and $q = 30$, the spectrums are block diagonal, and can be decomposed in 3 and 2 sectors respectively:

$$\mathcal{S}_{p,12} = \mathcal{S}_{p,12}^{(1)} \oplus \mathcal{S}_{p,12}^{(\sigma)} \oplus \mathcal{S}_{p,12}^{(\epsilon)} \quad , \quad \mathcal{S}_{p,30} = \mathcal{S}_{p,30}^{(1)} \oplus \mathcal{S}_{p,30}^{(\phi)} \ . \tag{1.6}$$

These block-diagonal structures are manifestations of extended symmetries, which constrain the fusion rules and correlation functions of these models, see Section 2.4.

Let us now introduce a more economical notation for the spectrums, which will make the analogy with loop models manifest. So far, we have only used integer Kac indices, and as a result each irreducible representation $\mathcal{R}_{r,s} \otimes \bar{\mathcal{R}}_{\bar{r},s}$ depends on 3 indices $(r, \bar{r}, s)$. However, we may write any pair of left and right dimensions as $(\Delta, \bar{\Delta}) = (\Delta_{r,s}, \Delta_{r,-s})$, provided we allow fractional indices $r, s$ [2]. We call $V_{(r,s)}$ the corresponding non-diagonal primary field, whose conformal spin is $\bar{\Delta} - \Delta = rs \in \mathbb{Z}$. For $r, s \in \mathbb{C}$, this primary field would belong to a Verma module of the product of the 2 Virasoro algebras, $V_{(r,s)} \in \mathcal{V}_{\Delta_{r,s}} \otimes \bar{\mathcal{V}}_{\Delta_{r,-s}}$. In our case, the values of the Kac indices such that our primary field belongs to our irreducible representation are

$$V_{\left(\frac{|r-\bar{r}|}{2}, \left(s - \frac{p(r+\bar{r})}{2q}\right)\text{sign}(r-\bar{r})\right)} \in \mathcal{R}_{r,s} \otimes \bar{\mathcal{R}}_{\bar{r},s} \ . \tag{1.7}$$

The resulting fields $V_{(r,s)}$ obey $r \in \mathbb{N}^*$ and $rs, qs \in \mathbb{Z}$. For completeness let us also write the reciprocal relation, although it is a bit tricky. Using an integer $a \in \mathbb{Z}$ such that $ap \equiv 1 \bmod q$, and writing $\{as\} = as - \lfloor as \rfloor$ the fractional part of $as \in \mathbb{Q}$, we have

$$V_{(r,s)} \in \mathcal{R}_{q\{as\}-r, p\{as\}-s} \otimes \bar{\mathcal{R}}_{q\{as\}+r, p\{as\}-s} . \tag{1.8}$$

Using also the notation $V^d_{\langle r,s \rangle} \in |\mathcal{R}_{r,s}|^2$ for a diagonal field of dimensions $\Delta = \bar{\Delta} = \Delta_{r,s}$, the spectrums (1.5) are equivalent to the lists of primary fields

$$\mathcal{P}_{p,12} = \left\{ V^d_{\langle r,s \rangle} \right\}_{\substack{r \in \{1,4,5\} \\ s \in ]0,p[}} \bigcup \left\{ V_{(2,s)} \right\}_{s \in ]-\frac{p}{2}, \frac{p}{2}[} \bigcup \left\{ V_{(3,s)} \right\}_{s \in \pm]-\frac{p}{3}, \frac{2p}{3}[} , \tag{1.9a}$$

$$\mathcal{P}_{p,18} = \left\{ V^d_{\langle r,s \rangle} \right\}_{\substack{r \in \{1,5,7,9\} \\ s \in ]0,p[}} \bigcup \left\{ V_{(r,s)} \right\}_{\substack{r \in \{2,4,8\} \\ s \in ]-\frac{p}{2}, \frac{p}{2}[}} \bigcup \left\{ V_{(3,s)} \right\}_{s \in \pm]-\frac{p}{3}, \frac{2p}{3}[} , \tag{1.9b}$$

$$\mathcal{P}_{p,30} = \left\{ V^d_{\langle r,s \rangle} \right\}_{\substack{r \in \{1,7,11,13\} \\ s \in ]0,p[}} \bigcup \left\{ V_{(2,s)} \right\}_{s \in ]-\frac{p}{2}, \frac{p}{2}[} \bigcup \left\{ V_{(r,s)} \right\}_{\substack{r \in \{3,9\} \\ s \in \pm]-\frac{p}{3}, \frac{2p}{3}[}}$$

$$\bigcup \left\{ V_{(5,s)} \right\}_{s \in \pm]-\frac{p}{5}, \frac{4p}{5}[} \bigcup \left\{ V_{(5,s)} \right\}_{s \in \pm]-\frac{2p}{5}, \frac{3p}{5}[} . \tag{1.9c}$$

Let us compare with spectrums of loop CFTs. For definiteness, we write the primary fields of the $PSU(n)$ CFT [6]:

$$\mathcal{P}_{PSU(n)} = \left\{ V^d_{\langle 1,s \rangle} \right\}_{s \in \mathbb{N}^*} \bigcup \left\{ V_{(r,s)} \right\}_{\substack{r \in \mathbb{N}^* \\ s \in \frac{1}{r}\mathbb{Z}}} , \tag{1.10}$$

The non-diagonal sectors of E-series minimal models are subsets of the non-diagonal sector of the $PSU(n)$ CFT. The main difference is that the second index of $V_{(r,s)}$ is a fraction whose denominator is a prime factor of $q$, and therefore bounded. This suggests that there could be generalizations $\widetilde{E}_{p,q}$ of E-series minimal models to generic values of $q$, such that the denominator of the second index increases with $q$. In the limit $p, q \to \infty$ with $\frac{p}{q} \to \beta_0^2$, the spectrum would tend to $\mathcal{P}_{PSU(n)}$. In particular, in the diagonal sector, the gap above $r = 1$ for $V^d_{\langle r,s \rangle}$ would increase to infinity. Therefore, we would expect $\lim_{\frac{p}{q} \to \beta_0^2} \widetilde{E}_{p,q}$ to be the $PSU(n)$ CFT.

The generalizations $\widetilde{E}_{p,q}$ would have to break rationality, irreducibility and/or modular invariance. In loop models, the fields $V_{(r,s)}$ with $s \in \mathbb{Z}^*$ are logarithmic at generic central charge [7]: $\widetilde{E}_{p,q}$ might well include such fields, which would break irreducibility and most probably lead to non-rationality as well. However, we have not been able to find new CFTs of that type. The numerical bootstrap methods of Section 1.3 can be used for testing the validity of conjectured spectrums, by looking for crossing-symmetric 4-point functions. But as is apparent in our notations (1.9), there are many plausible generalizations of E-series spectrums. We have unsuccessfully tested some of these generalizations in the cases $q = 9, 15$.

## 1.3 Bootstrapping correlation functions

Minimal models are the simplest and most well-known exactly solvable CFTs, and are commonly used for testing numerical bootstrap methods (or other techniques or ideas) before applying them to more challenging cases. But in this work we numerically bootstrap E-series minimal models in order to actually solve them. This is not as unreasonable as it seems, because these models have not been solved, in the sense that no explicit formula for 3-point structure constants is known.

The analytic bootstrap method [2] studies correlation functions involving the diagonal degenerate fields $V^d_{\langle 2,1 \rangle}$ and $V^d_{\langle 1,2 \rangle}$, and deduces how structure constants for $V_{(r,s)}$ or $V^d_{\langle r,s \rangle}$

behave under $r \to r + 2$ or $s \to s + 2$. However, in sharp contrast with the A-series and D-series models, E-series minimal models have spectrums that include $V_{\langle 1,2 \rangle}^d$ but not $V_{\langle 2,1 \rangle}^d$, and that are correspondingly invariant under $s \to s + 2$ (within the interval $s \in {]}0, p{[}$) but not $r \to r + 2$. So we cannot solve them by this method.

On the other hand, it is possible to use the same numerical bootstrap methods as in loop CFTs, whose main requirement is an exact knowledge of the spectrum [2]. Already in loop CFTs, it is then possible to infer exact results from numerical data. This is done by comparing numerical data with a universal ansatz, assuming that the ansatz only misses a polynomial factor [8]. The same ansatz will work for E-series minimal models. Nevertheless, adapting the method to loop models is technically challenging, and we will fully solve the $E_{p,12}$ models but not the $E_{p,18}$ or $E_{p,30}$ models. Let us explain why:

- The crucial parameter for the numerical bootstrap is the size of the spectrum, i.e. the number of primary fields = the number of irreducible representations. In minimal models, this number is finite but not particularly small, as it is of the order $\frac{1}{2}pq$, see Table (1.2).

- Virasoro conformal blocks are numerically computed using Zamolodchikov's recursive representation. In this representation, blocks are written as sums of terms that have poles at rational values of the central charge $c$, even though the blocks themselves are smooth. In loop models, we can take complex values of $c$, far away from all poles. For minimal models, we have to take values of $c$ that differ from the desired rational values (1.1) by small amounts. The calculations therefore have one more numerical parameter, and one more potential source of systematic errors and numerical instability.

- Inferring exact results from numerical data is a tedious task, done one a case-by-case basis. In the $E_{p,12}$ model we find 13 nonzero 3-point structure constants that are independent modulo symmetries. This number should scale as $O(q^3)$, and we expect $\sim 44$ structure constants in the $E_{p,18}$ case and $\sim 203$ structure constants in the $E_{p,30}$ case.

We have used existing Python code in the public repository [9] for bootstrapping $E_{p,12}$ models. In that repository we have added a notebook for computing structure constants and checking crossing symmetry in $E_{p,12}$ models. The notebook also computes fusion products in all E-series models.

## 1.4  Chiral and non-chiral fusion rules

In order to interpret our numerical results on 3-point structure constants, we will use the concept of **non-chiral fusion rules**, which we now define after recalling the more familiar chiral fusion rules:

- **Chiral fusion rules** describe the fusion product of representations of the chiral algebra. Given 2 representations $\mathcal{R}_i, \mathcal{R}_j$, we write

$$\mathcal{R}_i \times \mathcal{R}_j \simeq \bigoplus_k N_{i,j}^k \mathcal{R}_k \iff N_{i,j}^k = \dim \mathrm{Hom}\left(\mathcal{R}_i \times \mathcal{R}_j, \mathcal{R}_k\right) . \tag{1.11}$$

- **Non-chiral fusion rules** describe the fields that appear in operator product expansions. If we write a primary field as $V_I = V_{(i,\bar{i})} \in \mathcal{R}_i \otimes \bar{\mathcal{R}}_{\bar{i}}$, the OPE of 2 primary

fields reads

$$V_I V_J \sim \sum_K \sum_{\alpha=1}^{M_{I,J}^K} C_{I,J}^{K,(\alpha)} \left( V_K + \text{descendants}_{(\alpha)} \right) \quad \text{with} \quad C_{I,J}^{K,(\alpha)} \neq 0 \;, \qquad (1.12)$$

then we define the non-chiral fusion rules as

$$V_I * V_J = \sum_K M_{I,J}^K V_K \;. \qquad (1.13)$$

We assume that the representations $\mathcal{R}_i$ belong to a category where the fusion product is well-defined and associative. Then associativity implies $\sum_\ell N_{i,j}^\ell N_{\ell,k}^m = \sum_\ell N_{j,k}^\ell N_{i,\ell}^m$. In the case of the Virasoro algebra, fusion multiplicities are trivial, i.e. $N_{i,j}^k \in \{0,1\}$. For W-algebras the fusion multiplicities can be arbitrary integers, or even infinite. The coefficients $N_{i,j}^k$ obey further algebraic properties such as the Verlinde formula.

Chiral fusion constrains OPEs, leading to

$$M_{I,J}^K \leq N_{i,j}^k N_{\bar{i},\bar{j}}^{\bar{k}} \;, \qquad (1.14)$$

where we recall $I = (i, \bar{i})$. Indeed, if we had $N_{i,j}^k \geq 2$, then contributions of descendants would not be determined by primaries, as indicated in Eq. (1.12) by their dependence on $\alpha$. In particular, in E-series minimal models, we have $M_{I,J}^K \in \{0,1\}$.

Non-chiral fusion rules are a notation for indicating which primary fields have nonzero OPE coefficients. While OPEs are associative, the non-chiral fusion rules have no reason to be associative, or to obey nice algebraic properties such as the Verlinde formula. See Eq. (3.3) for an explicit example of non-associativity.

Nevertheless, non-chiral fusion rules can be associative when they involve **non-chiral simple currents**: primary fields whose OPE with any other primary field involves only one field. In particular, a simple current that squares to the identity field $V_0$ induces a $\mathbb{Z}_2$ symmetry of the non-chiral fusion rules. Indeed, let $V_s$ be such that $V_s * V_I = V_{\sigma(I)}$ and $V_s * V_s = V_0$ with $V_0 * V_I = V_I$. Associativity of fusion products that involve $V_s$ implies that $\sigma$ is an involution, and

$$M_{I,J}^K = M_{\sigma(I),\sigma(J)}^K = M_{I,\sigma(J)}^{\sigma(K)} = M_{\sigma(I),J}^{\sigma(K)} \;. \qquad (1.15)$$

We will find non-trivial simple currents in the $E_{p,18}$ and $E_{p,30}$ minimal models.

Non-chiral fusion rules are useful for studying E-series minimal models, for the following reasons:

- From numerical bootstrap results, it can be hard to determine exact values of structure constants, but it is much easier to see whether they vanish or not. So non-chiral fusion rules are non-trivial results that can be useful although they fall short of a full solution of the model.

- Conversely, using non-chiral fusion rules can help solve crossing symmetry equations, by reducing the number of terms beyond what chiral symmetry requires.

- When they vanish, structure constants usually have good reasons for doing so. In particular, non-chiral fusion rules reflect symmetries of the CFT.

- Any subset of fields that is closed under non-chiral fusion rules defines a submodel: a rational CFT that is consistent on the sphere but not on the torus. Submodels of unitary minimal models have been classified, and in the $E_{p,12}$ case the classification can be rederived from the fusion rules [10].

Our definition of non-chiral fusion rules has already appeared in the literature, but has not seen much use [11]. Quite understandably, non-chiral fusion rules have more often be written in special cases where they are associative. These cases include algebras of simple currents, for example algebras of anyons [12], or algebras of primary fields in free theories [13]. Moreover, in diagonal CFTs, non-chiral fusion rules coincide with chiral fusion rules, and are therefore associative.

# 2 Constraints on 3-point structure constants

Let us review the known constraints on 3-point structure constants. In Sections 3 and 4 we will deduce their detailed consequences in E-series minimal models.

We use the following notations for 2-point and 3-point structure constants, and for OPE coefficients:

$$\langle V_i V_j \rangle \propto \delta_{i,j} B_i \quad , \quad \langle V_i V_j V_k \rangle \propto C_{i,j,k} \quad , \quad V_i V_j \sim \sum_k C_{i,j}^k V_k \; . \tag{2.1}$$

These objects obey the relations

$$B_i = C_{i,i,1} \quad , \quad C_{i,j,k} = C_{i,j}^k B_k \; , \tag{2.2}$$

where the index 1 refers to the identity field $V_{\langle 1,1 \rangle}^d$. From 2-point and 3-point structure constants, we can assemble 4-point structure constants, which appear in decompositions of 4-point functions into conformal blocks. For example, the decomposition into $s$-channel blocks $\mathcal{G}_k^{(s)}$ reads

$$\left\langle V_1(z) V_2(0) V_3(\infty) V_4(1) \right\rangle = \sum_k D_k^{(s)} \mathcal{G}_k^{(s)}(z) \quad \text{with} \quad D_k^{(s)} = \frac{C_{1,2,k} C_{k,3,4}}{B_k} \; , \tag{2.3}$$

where we have used the global conformal invariance to fix $\{z_2, z_3, z_4\} = \{0, \infty, 1\}$.

## 2.1 Null vectors

First, 3-point structure constants are constrained by chiral fusion rules, which follow from conformal symmetry and the existence of null vectors. At a central charge $c = c_{p,q}$, the chiral fusion rules of irreducible, fully degenerate representations of the Virasoro algebra are [14]

$$\mathcal{R}_{r_1,s_1} \times \mathcal{R}_{r_2,s_2} = \bigoplus_{r_3 \overset{2}{=} |r_1-r_2|+1}^{\min(r_1+r_2, 2q-r_1-r_2)-1} \bigoplus_{s_3 \overset{2}{=} |s_1-s_2|+1}^{\min(s_1+s_2, 2p-s_1-s_2)-1} \mathcal{R}_{r_3,s_3} \; , \tag{2.4}$$

where $(r_i, s_i) \in ]0, q[ \times ]0, p[$, and the sums run by increments of 2. We denote $N_{\langle r_1,s_1 \rangle \langle r_2,s_2 \rangle}^{\langle r_3,s_3 \rangle} = N_{\langle r_1,s_1 \rangle \langle r_2,s_2 \rangle \langle r_3,s_3 \rangle}$ the chiral fusion coefficients, which are symmetric under permutations of the 3 representations. Due to the ambiguity $\mathcal{R}_{r,s} = \mathcal{R}_{q-r,p-s}$, there are 2 cases where $N_{\langle r_1,s_1 \rangle \langle r_2,s_2 \rangle \langle r_3,s_3 \rangle} \neq 0$:

| Odd case | Even case |
|---|---|
| $2q - r_1 - r_2 - r_3 \in 1 + 2\mathbb{N}$ | $r_1 + r_2 + r_3 - q \in 1 + 2\mathbb{N}$ |
| $r_i + r_j - r_k \in 1 + 2\mathbb{N}$ | $q + r_k - r_i - r_j \in 1 + 2\mathbb{N}$ |
| $2p - s_1 - s_2 - s_3 \in 1 + 2\mathbb{N}$ | $s_1 + s_2 + s_3 - p \in 1 + 2\mathbb{N}$ |
| $s_i + s_j - s_k \in 1 + 2\mathbb{N}$ | $p + s_k - s_i - s_j \in 1 + 2\mathbb{N}$ |

$$\tag{2.5}$$

In E-series minimal models, $q$ is even while $p$ is odd. Therefore, the integers $r_1 \pm r_2 \pm r_3$ are always odd, while the integers $s_1 \pm s_2 \pm s_3$ can be even or odd, and we name the 2 cases after its parity. After lifting the ambiguity $\mathcal{R}_{r,s} = \mathcal{R}_{q-r,p-s}$ by choosing one value of $r$ in each pair $\{r, q-r\}$, being even or odd will be a well-defined property of any 3-point coupling.

For example, in the case $q = 12$, we have the following chiral fusion rules:

$$\mathcal{R}_{7,s_1} \times \mathcal{R}_{7,s_2} = \bigoplus_{s_3 \overset{2}{=} |s_1-s_2|+1}^{\min(s_1+s_2, 2p-s_1-s_1)-1} \left( \mathcal{R}_{1,s_3} \oplus \mathcal{R}_{3,s_3} \oplus \mathcal{R}_{5,s_3} \oplus \mathcal{R}_{7,s_3} \oplus \mathcal{R}_{9,s_3} \right) . \quad (2.6)$$

Then $N_{\langle 7,s_1 \rangle \langle 7,s_2 \rangle \langle 1,s_3 \rangle}$ is nonzero only in the odd case. On the other hand, $N_{\langle 7,s_1 \rangle \langle 7,s_2 \rangle \langle 7,s_3 \rangle}$ is nonzero in both odd and even cases, due to the term $\mathcal{R}_{5,s_3} = \mathcal{R}_{7,p-s_3}$, and the situation is the same for $N_{\langle 7,s_1 \rangle \langle 7,s_2 \rangle \langle 3,s_3 \rangle}$. Explicitly, the case $s_1 = s_2 = 1$ reads

$$\mathcal{R}_{7,1} \times \mathcal{R}_{7,1} = \mathcal{R}_{1,1} \oplus (\mathcal{R}_{3,1} \oplus \mathcal{R}_{3,p-1}) \oplus (\mathcal{R}_{7,1} \oplus \mathcal{R}_{7,p-1}) . \quad (2.7)$$

In the 3-dimensional space with coordinates $r_i - \frac{q}{2}$, odd couplings belong to the tetrahedron with vertices $\left\{ \frac{q}{2}(\epsilon_1, \epsilon_2, \epsilon_3) \right\}_{\substack{\epsilon_i \in \{\pm 1\} \\ \epsilon_1 \epsilon_2 \epsilon_3 = -1}}$, and even couplings to the tetrahedron with the vertices $\left\{ \frac{q}{2}(\epsilon_1, \epsilon_2, \epsilon_3) \right\}_{\substack{\epsilon_i \in \{\pm 1\} \\ \epsilon_1 \epsilon_2 \epsilon_3 = 1}}$. The intersection of the two tetrahedrons is the octahedron with vertices $\left\{ (\pm \frac{q}{2}, 0, 0), (0, \pm \frac{q}{2}, 0), (0, 0, \pm \frac{q}{2}) \right\}$. The intersection of the octant $\{ r_i - \frac{q}{2} \leq 0 \}$ with the even tetrahedron is included in the odd tetrahedron. In the E-series minimal models, we will choose $r \leq \frac{q}{2}$, except in the case $q = 12$ where we allow $r = 7$. As a result, any allowed even coupling will also come with an allowed odd coupling. In other words, for any $(r_1, r_2, r_3)$, we have either no couplings at all, or odd couplings, or even and odd couplings, with even couplings appearing if $r_1 + r_2 + r_3 > q$.

## 2.2 Interchiral symmetry

In all E-series minimal models, there exists a degenerate diagonal field $V_{\langle 1,2 \rangle}^d$. The associativity of OPEs that involve this field implies constraints on structure constants. We may view these constraints as manifestations of a symmetry algebra, called an interchiral algebra [2].

The field $V_{\langle 1,2 \rangle}^d$ transforms in the representation $\mathcal{R}_{1,2} \otimes \bar{\mathcal{R}}_{1,2}$. The chiral fusion rules of the Virasoro representation $\mathcal{R}_{1,2}$ are

$$\mathcal{R}_{1,2} \times \mathcal{R}_{r,s} = \delta_{s>1} \mathcal{R}_{r,s-1} \oplus \delta_{s<p-1} \mathcal{R}_{r,s+1} . \quad (2.8)$$

We assume that in E-series minimal models, the non-chiral fusion rules of $V_{\langle 1,2 \rangle}^d$ follow from the chiral fusion rules of $\mathcal{R}_{1,2}$. In other words, in the OPE of $V_{\langle 1,2 \rangle}^d$ with another primary field, we obtain all primary fields that are allowed by chiral fusion and belong to the spectrum. The resulting non-chiral fusion rules are easy enough to write if we parametrize fields by integer left and right Kac indices,

$$V_{\langle 1,2 \rangle}^d * V_{(r,s)(\bar{r},s)} = \delta_{s>1} V_{(r,s-1)(\bar{r},s-1)} + \delta_{s<p-1} V_{(r,s+1)(\bar{r},s+1)} , \quad (2.9)$$

including, in the diagonal case $r = \bar{r}$,

$$V_{\langle 1,2 \rangle}^d * V_{\langle r,s \rangle}^d = \delta_{s>1} V_{\langle r,s-1 \rangle}^d + \delta_{s<p-1} V_{\langle r,s+1 \rangle}^d . \quad (2.10)$$

Things get a bit trickier with the notation $V_{(r,s)}$ for non-diagonal fields, because the range of $s$ is no longer $]0, p[$, but some other interval of the same length, say $]s_0, s_0 + p[$. Then the non-chiral fusion rules are

$$V^d_{\langle 1,2 \rangle} * V_{(r,s)} = \delta_{s>s_0+1} V_{(r,s-1)} + \delta_{s<s_0+p-1} V_{(r,s+1)} \ . \tag{2.11}$$

This leads to constraints on the behaviour of 3-point structure constants under integer shifts of $s_i$. To be precise, we obtain equations that relate all nonzero structure constants of the type $C_{(r_1,s_1)(r_2,s_2)(r_3,s_3)}$ with a given parity $s_1 + s_2 + s_3$, where each pair of Kac indices $(r_i, s_i)$ may now belong to a diagonal or a non-diagonal field. Given indices $r_1, r_2, r_3$, we write $C^{\mathrm{odd}}_{r_1,r_2,r_3}$ and $C^{\mathrm{even}}_{r_1,r_2,r_3}$ for the independent structure constants, when they exist. And in fusion rules, we may omit the second Kac index, while indicating parity constraints by writing a superscript $^e$ when both even and odd parities are allowed, whereas by default only odd couplings appear. In the $E_{p,30}$ models there are cases where only even couplings appear, which we will indicate using a superscript $^E$. For example, in these notations, the chiral fusion rules (2.6) may be summarized as

$$\mathcal{R}_7 \times \mathcal{R}_7 = \mathcal{R}_1 \oplus {}^e \mathcal{R}_3 \oplus {}^e \mathcal{R}_7 \ . \tag{2.12}$$

## 2.3 Simple currents

We will now construct non-chiral simple currents from chiral simple currents. In principle there might exist non-chiral simple currents that are not built from chiral simple currents, but we will not find any in E-series minimal models.

By a chiral simple current we mean a representation whose fusion product with any other representation yields only 1 representation. In E-series minimal models, these are the representations $\mathcal{R}_{1,1}$ and $\mathcal{R}_{1,p-1}$. Their fusion rules are

$$\mathcal{R}_{1,1} \times \mathcal{R}_{r,s} = \mathcal{R}_{r,s} \quad , \quad \mathcal{R}_{1,p-1} \times \mathcal{R}_{r,s} = \mathcal{R}_{r,p-s} \ . \tag{2.13}$$

From these chiral simple currents, we can build the diagonal simple current $V^d_{\langle 1,1 \rangle}$, which is trivial. However, we can also build the non-diagonal simple current $V_{(1,1)(1,p-1)} = V_{(1,1)(q-1,1)} = V_{(\frac{q}{2}-1,\frac{p}{2}-1)}$, which is non-trivial if it exists. This field does not exist in the $E_{p,12}$ minimal models, in fact its conformal spin would be half-integer. But it exists in the $E_{p,18}$ and $E_{p,30}$ cases, under the names $V_{(8,\frac{p}{2}-1)}$ and $V_{(14,\frac{p}{2}-1)}$ respectively.

It is not hard to write the non-chiral fusion rules of the simple current $V_{(\frac{q}{2}-1,\frac{p}{2}-1)}$ with itself and with diagonal fields,

$$V_{(\frac{q}{2}-1,\frac{p}{2}-1)} * V_{(\frac{q}{2}-1,\frac{p}{2}-1)} = V^d_{\langle 1,1 \rangle} \ , \tag{2.14}$$

$$V_{(\frac{q}{2}-1,\frac{p}{2}-1)} * V^d_{\langle r,s \rangle} \underset{r<\frac{q}{2}}{=} V_{(\frac{q}{2}-r,\frac{p}{2}-s)} \ , \tag{2.15}$$

$$V_{(\frac{q}{2}-1,\frac{p}{2}-1)} * V^d_{\langle \frac{q}{2},s \rangle} = V^d_{\langle \frac{q}{2},s \rangle} \ . \tag{2.16}$$

Fusion with non-diagonal fields is a bit trickier to write in general, and we will work it out explicitly in each case.

## 2.4 Extended symmetries

The block diagonal structures of the $E_{p,12}$ and $E_{p,30}$ spectrums (1.6) suggest the existence of extended symmetries, beyond the Virasoro symmetry that is assumed by definition of minimal models. And these extended symmetries constrain non-chiral fusion rules [14, Exercises 10.14 to 10.16].

In the case of $E_{p,12}$ minimal models, the extended symmetry is described by an Ising-type category. The objects are the blocks $\mathbf{1}, \sigma, \epsilon$ that appear in Eq. (1.6), and the products are

$$\mathbf{1} \times \mathbf{1} = \mathbf{1} \quad , \quad \mathbf{1} \times \sigma = \sigma \quad , \quad \mathbf{1} \times \epsilon = \epsilon \quad ,$$
$$\epsilon \times \epsilon = \mathbf{1} \quad , \quad \sigma \times \sigma = \mathbf{1} + \epsilon \quad , \quad \epsilon \times \sigma = \sigma \ . \tag{2.17}$$

It turns out that almost all these constraints already follow from the chiral fusion rules. The one exception is

$$C_{7,7,7}^{\text{even}} = 0 \ . \tag{2.18}$$

In the spectrum (1.5a), the fields $V_{\langle 7,s \rangle}^{d}$ with $s$ odd belong to the identity sector of the Ising-type category, whereas $V_{\langle 7,s \rangle}^{d}$ with $s$ even belongs to the $\epsilon$ sector, under the name $V_{\langle 5,p-s \rangle}^{d}$. The vanishing of the coupling $\langle \mathbf{1}\mathbf{1}\epsilon \rangle$ therefore leads to the vanishing of $C_{7,7,7}^{\text{even}}$.

Similarly, in the case of $E_{p,30}$, the extended symmetry is described by a Lee–Yang-type category, with the fusion rules

$$\mathbf{1} \times \mathbf{1} = \mathbf{1} \quad , \quad \mathbf{1} \times \phi = \phi \quad , \quad \phi \times \phi = \mathbf{1} + \phi \ . \tag{2.19}$$

Here, it is the vanishing of the coupling $\langle \mathbf{1}\mathbf{1}\phi \rangle$ that constrains non-chiral fusion rules. To get constraints that do not already follow from chiral fusion rules, we need fields in the $\mathbf{1}$ sector of the type $V_{(r,s)(\bar{r},s)}$ with $r, \bar{r} \notin \{1, 29\}$, i.e. fields that are not built from a left or right chiral simple current. Such fields are $V_{\langle 11,s \rangle}^{d}$ or $V_{(4,s)}$, but these two cases are related by fusion with the non-chiral simple current. Therefore, modulo the $\mathbb{Z}_2$ simple current symmetry, all constraints boil down to

$$\forall X \in \phi \ , \quad C_{11,11,X} = 0 \ . \tag{2.20}$$

## 2.5 Permutations

Under odd permutations of the 3 fields, any 3-point structure constant picks a sign $(-1)^{S_1+S_2+S_3}$, where $S_i = \Delta_i - \bar{\Delta}_i$ is the conformal spin. Therefore,

$$S_Y \in 2\mathbb{Z} + 1 \quad \implies \quad \forall X \ , \quad C_{X,X,Y} = 0 \ . \tag{2.21}$$

In particular, in E-series minimal models, all non-diagonal fields of the type $V_{(2,s)}$ have odd spins, therefore $C_{X,X,2} = 0$. Similarly, in $E_{p,30}$ minimal models, we have $C_{X,X,14} = 0$.

In the case of a non-diagonal field $V_{(r,s)}$ with $r$ odd, the parity of the spin changes under $s \to s + 1$. Therefore, for any field $X$, only one of the two couplings $C_{X,X,r}^{\text{even}}$ or $C_{X,X,r}^{\text{odd}}$ can be nonzero. The coupling that is allowed by permutation symmetry may or may not be allowed by chiral fusion rules: we will see examples of both cases.

## 2.6 Parity

Crossing symmetry equations are invariant under complex conjugation of positions, and therefore equivalently under the exchange of left and right conformal dimensions. For a non-diagonal field, this exchange amounts to $V_{(r,s)} \to V_{(r,-s)}$. We will use the compact notation $X \to \bar{X}$ for this operation, with $X = \bar{X}$ for diagonal fields.

This does not immediately imply that all structure constants are invariant under flipping the signs of $s$ for all non-diagonal fields. In loop models, there indeed exist parity-odd solutions of crossing symmetry [8]. However, in E-series minimal models, we only find parity-even solutions. We therefore assume that all structure constants are parity-even,

$$C_{X,Y,Z} = C_{\bar{X},\bar{Y},\bar{Z}} \ . \tag{2.22}$$

# 3 Non-chiral fusion rules

Before writing the fusion rules explicitly, let us summarize which constraints apply in $E_{p,q}$ minimal models in each one of the cases $q = 12, 18, 30$. In the following table, we indicate constraints that apply in green, while omitting constraints that are redundant with previous constraints:

| Constraints | $E_{p,12}$ | $E_{p,18}$ | $E_{p,30}$ |
|---:|:---:|:---:|:---:|
| Null vectors | 🟩 | 🟩 | 🟩 |
| Interchiral symmetry | 🟩 | 🟩 | 🟩 |
| Simple currents | | 🟩 | 🟩 |
| Extended symmetry | 🟩 | | 🟩 |
| Permutations | | 🟩 | 🟩 |
| Parity | 🟩 | 🟩 | 🟩 |
| Unexplained | | | 🟩 |

## 3.1 Case $q = 12$

For each field we give a compact notation $D_k$ (diagonal field) or $N_k$ (non-diagonal field), the full name of the field, the range of values of the second Kac index, and the values of the integer left and right Kac indices.

| Notation | Field | Values of $s$ | Left | Right | |
|:---:|:---:|:---:|:---:|:---:|:---:|
| $D_1$ | $V^d_{\langle 1,s \rangle}$ | $]0, p[$ | $(1, s)$ | $(1, s)$ | |
| $D_4$ | $V^d_{\langle 4,s \rangle}$ | $]0, p[$ | $(4, s)$ | $(4, s)$ | |
| $D_7$ | $V^d_{\langle 7,s \rangle}$ | $]0, p[$ | $(7, s)$ | $(7, s)$ | (3.1) |
| $N_2$ | $V_{(2,s)}$ | $]-\frac{p}{2}, \frac{p}{2}[$ | $(4, \frac{p}{2} - s)$ | $(8, \frac{p}{2} - s)$ | |
| $N_3$ | $V_{(3,s)}$ | $]-\frac{2p}{3}, \frac{p}{3}[$ | $(1, \frac{p}{3} - s)$ | $(7, \frac{p}{3} - s)$ | |
| $\bar{N}_3$ | $V_{(3,s)}$ | $]-\frac{p}{3}, \frac{2p}{3}[$ | $(7, \frac{p}{3} + s)$ | $(1, \frac{p}{3} + s)$ | |

We now write the fusion rules in our compact notation, which omits the second Kac index. The dependence on that index can be recovered as explained in Section 2.2. In particular, the superscript $^e$ appears when that index takes odd and even values, as opposed to having

a fixed parity:

$$D_1 * X = X \quad (\forall X) \,, \tag{3.2a}$$

$$D_4 * D_4 = D_1 + {}^e D_7 + N_3 + \bar{N}_3 \,, \tag{3.2b}$$

$$D_4 * D_7 = {}^e D_4 + {}^e N_2 \,, \tag{3.2c}$$

$$D_7 * D_7 = D_1 + D_7 + N_3 + \bar{N}_3 \,, \tag{3.2d}$$

$$D_4 * N_2 = {}^e D_7 + N_3 + \bar{N}_3 \,, \tag{3.2e}$$

$$D_4 * N_3 = D_4 + N_2 \,, \tag{3.2f}$$

$$D_7 * N_2 = {}^e D_4 + {}^e N_2 \,, \tag{3.2g}$$

$$D_7 * N_3 = D_7 + \bar{N}_3 \,, \tag{3.2h}$$

$$N_2 * N_2 = D_1 + {}^e D_7 + N_3 + \bar{N}_3 \,, \tag{3.2i}$$

$$N_2 * N_3 = D_4 + N_2 \,, \tag{3.2j}$$

$$N_3 * N_3 = D_1 + N_3 \,, \tag{3.2k}$$

$$N_3 * \bar{N}_3 = D_7 \,. \tag{3.2l}$$

We omit rules that can be deduced by parity, for example $D_7 * \bar{N}_3 = D_7 + N_3$. Modulo the interchiral and parity symmetries, these rules are dictated by null vectors, together with the constraint (2.18) from the extended symmetry which leads to the appearance of only $D_7$ rather than ${}^e D_7$ in Eq. (3.2d). Permutation symmetry does not add more constraints: for example, $C_{2,2,3}^{\text{even}} = 0$ (dictated by null vectors) implies that $\langle V_{(2,s)} V_{(2,s)} V_{(3,s')} \rangle = 0$ if $\frac{p}{3} - s'$ is even, equivalently if $V_{(3,s')}$ has an odd conformal spin $3s'$ (as required by permutation symmetry).

As explained in Section 1.4, non-chiral fusion rules have no reason to be associative. For example,

$$D_4 * (N_3 * N_3) = D_4 * (D_1 + N_3) = 2D_4 + N_2 \,, \tag{3.3a}$$

$$(D_4 * N_3) * N_3 = (D_4 + N_2) * N_3 = 2D_4 + 2N_2 \,. \tag{3.3b}$$

## 3.2 Case $q = 18$

| Notation | Field | Values of $s$ | Left | Right |
|----------|-------|---------------|------|-------|
| $D_1$ | $V_{\langle 1,s \rangle}^d$ | $]0, p[$ | $(1, s)$ | $(1, s)$ |
| $D_5$ | $V_{\langle 5,s \rangle}^d$ | $]0, p[$ | $(5, s)$ | $(5, s)$ |
| $D_7$ | $V_{\langle 7,s \rangle}^d$ | $]0, p[$ | $(7, s)$ | $(7, s)$ |
| $D_9$ | $V_{\langle 9,s \rangle}^d$ | $]0, p[$ | $(9, s)$ | $(9, s)$ |
| $N_2$ | $V_{(2,s)}$ | $]-\frac{p}{2}, \frac{p}{2}[$ | $(7, \frac{p}{2} - s)$ | $(11, \frac{p}{2} - s)$ |
| $N_4$ | $V_{(4,s)}$ | $]-\frac{p}{2}, \frac{p}{2}[$ | $(5, \frac{p}{2} - s)$ | $(13, \frac{p}{2} - s)$ |
| $N_8$ | $V_{(8,s)}$ | $]-\frac{p}{2}, \frac{p}{2}[$ | $(1, \frac{p}{2} - s)$ | $(17, \frac{p}{2} - s)$ |
| $N_3$ | $V_{(3,s)}$ | $]-\frac{2p}{3}, \frac{p}{3}[$ | $(3, \frac{p}{3} - s)$ | $(9, \frac{p}{3} - s)$ |
| $\bar{N}_3$ | $V_{(3,s)}$ | $]-\frac{p}{3}, \frac{2p}{3}[$ | $(9, \frac{p}{3} + s)$ | $(3, \frac{p}{3} + s)$ |

$$\tag{3.4}$$

The distinction between even and odd couplings arises from solving the ambiguity $\mathcal{R}_{r,s} = \mathcal{R}_{q-r,p-s}$ by choosing a value in $\{r, q - r\}$, but this is not possible for $D_9$ as $r = q - r$ in

this case. As a result, whenever $D_9$ is involved, there is no distinction between odd and even couplings.

The simple current $N_8$ generates a $\mathbb{Z}_2$ symmetry with the orbits

$$\{D_9\}, \{N_3\}, \{\bar{N}_3\}, \{D_1, N_8\}, \{D_5, N_4\}, \{D_7, N_2\} . \tag{3.5}$$

This means that we have the fusion rules

$$N_8 * D_9 = D_9 \quad , \quad N_8 * D_5 = N_4 \quad , \quad N_8 * N_4 = D_5 \quad , \quad \text{etc.} \tag{3.6}$$

Modulo this simple current symmetry, and modulo parity, the non-chiral fusion rules are

$$D_1 * X = X \quad (\forall X) , \tag{3.7a}$$
$$D_5 * D_5 = D_1 + D_5 + D_7 + D_9 + N_3 + \bar{N}_3 , \tag{3.7b}$$
$$D_5 * D_7 = D_5 + {}^e D_7 + D_9 + {}^e N_2 + N_3 + \bar{N}_3 , \tag{3.7c}$$
$$D_5 * D_9 = D_5 + D_7 + D_9 + N_2 + N_4 , \tag{3.7d}$$
$$D_5 * N_3 = D_5 + D_7 + N_2 + N_3 + N_4 , \tag{3.7e}$$
$$D_7 * D_7 = D_1 + {}^e D_5 + {}^e D_7 + D_9 + N_3 + \bar{N}_3 + {}^e N_4 , \tag{3.7f}$$
$$D_7 * D_9 = D_5 + D_7 + N_2 + N_3 + \bar{N}_3 + N_4 , \tag{3.7g}$$
$$D_7 * N_3 = D_5 + D_7 + D_9 + N_2 + {}^e \bar{N}_3 + N_4 , \tag{3.7h}$$
$$D_9 * D_9 = D_1 + D_5 + D_9 + N_4 + N_8 , \tag{3.7i}$$
$$D_9 * N_3 = D_7 + N_2 + \bar{N}_3 , \tag{3.7j}$$
$$N_3 * N_3 = D_1 + D_5 + N_3 + N_4 + N_8 , \tag{3.7k}$$
$$N_3 * \bar{N}_3 = {}^e D_7 + D_9 + {}^e N_2 \ . \ . \tag{3.7l}$$

To be complete, let us write the remainder of the fusion rules, deduced using the relation (1.15) applied to the simple current symmetry (3.5):

$$D_5 * N_2 = {}^e D_7 + D_9 + {}^e N_2 + N_3 + \bar{N}_3 + N_4 , \tag{3.8a}$$
$$D_5 * N_4 = D_9 + N_2 + N_3 + \bar{N}_3 + N_4 + N_8 , \tag{3.8b}$$
$$D_7 * N_2 = {}^e D_5 + D_9 + {}^e N_2 + N_3 + \bar{N}_3 + {}^e N_4 + N_8 , \tag{3.8c}$$
$$D_7 * N_4 = {}^e D_7 + D_9 + {}^e N_2 + N_3 + \bar{N}_3 + N_4 , \tag{3.8d}$$
$$D_9 * N_2 = D_5 + D_7 + N_2 + N_3 + \bar{N}_3 + N_4 , \tag{3.8e}$$
$$D_9 * N_4 = D_5 + D_7 + D_9 + N_2 + N_4 , \tag{3.8f}$$
$$N_2 * N_2 = D_1 + {}^e D_5 + {}^e D_7 + D_9 + N_3 + \bar{N}_3 + {}^e N_4 , \tag{3.8g}$$
$$N_2 * N_3 = D_5 + D_7 + D_9 + N_2 + {}^e \bar{N}_3 + N_4 , \tag{3.8h}$$
$$N_2 * N_4 = D_5 + {}^e D_7 + D_9 + {}^e N_2 + N_3 + \bar{N}_3 , \tag{3.8i}$$
$$N_3 * N_4 = D_5 + D_7 + N_2 + N_3 + N_4 , \tag{3.8j}$$
$$N_4 * N_4 = D_1 + D_5 + D_7 + D_9 + N_3 + \bar{N}_3 \ . \tag{3.8k}$$

In these results, the vanishings $\langle D_7 D_7 N_2 \rangle = \langle D_9 D_9 N_2 \rangle = 0$ are due to permutation symmetry. From $\langle D_9 D_9 N_2 \rangle = 0$ we deduce $\langle D_9 D_9 D_7 \rangle = 0$ using simple current symmetry. Similarly, $\left\langle V^d_{\langle 9, s_1 \rangle} V^d_{\langle 9, s_1 \rangle} N_{(3, s_3)} \right\rangle$ vanishes by permutation symmetry if the spin of $N_{(3, s_3)}$ is odd. However, this coupling involves $D_9$, so there is no distinction between even and odd couplings. This implies that the couplings $\left\langle V^d_{\langle 9, s_1 \rangle} V^d_{\langle 9, s_2 \rangle} N_{(3, s_3)} \right\rangle$ for all values of $s_i$ are related by interchiral symmetry, therefore $\langle D_9 D_9 N_3 \rangle = 0$. We conclude that the non-chiral fusion rules are completely determined by the constraints of Section 2.

## 3.3 Case $q = 30$

| Notation | Field | Values of $s$ | Left | Right |
|:---:|:---:|:---:|:---:|:---:|
| $D_1$ | $V^d_{\langle 1,s\rangle}$ | $]0,p[$ | $(1,s)$ | $(1,s)$ |
| $D_7$ | $V^d_{\langle 7,s\rangle}$ | $]0,p[$ | $(7,s)$ | $(7,s)$ |
| $D_{11}$ | $V^d_{\langle 11,s\rangle}$ | $]0,p[$ | $(11,s)$ | $(11,s)$ |
| $D_{13}$ | $V^d_{\langle 13,s\rangle}$ | $]0,p[$ | $(13,s)$ | $(13,s)$ |
| $N_2$ | $V_{(2,s)}$ | $]-\frac{p}{2},\frac{p}{2}[$ | $(13,\frac{p}{2}-s)$ | $(17,\frac{p}{2}-s)$ |
| $N_4$ | $V_{(4,s)}$ | $]-\frac{p}{2},\frac{p}{2}[$ | $(11,\frac{p}{2}-s)$ | $(19,\frac{p}{2}-s)$ |
| $N_8$ | $V_{(8,s)}$ | $]-\frac{p}{2},\frac{p}{2}[$ | $(7,\frac{p}{2}-s)$ | $(23,\frac{p}{2}-s)$ |
| $N_{14}$ | $V_{(14,s)}$ | $]-\frac{p}{2},\frac{p}{2}[$ | $(1,\frac{p}{2}-s)$ | $(29,\frac{p}{2}-s)$ |
| $N_3$ | $V_{(3,s)}$ | $]-\frac{2p}{3},\frac{p}{3}[$ | $(7,\frac{p}{3}-s)$ | $(13,\frac{p}{3}-s)$ |
| $\bar{N}_3$ | $V_{(3,s)}$ | $]-\frac{p}{3},\frac{2p}{3}[$ | $(13,\frac{p}{3}+s)$ | $(7,\frac{p}{3}+s)$ |
| $N_9$ | $V_{(9,s)}$ | $]-\frac{2p}{3},\frac{p}{3}[$ | $(1,\frac{p}{3}-s)$ | $(19,\frac{p}{3}-s)$ |
| $\bar{N}_9$ | $V_{(9,s)}$ | $]-\frac{p}{3},\frac{2p}{3}[$ | $(19,\frac{p}{3}+s)$ | $(1,\frac{p}{3}+s)$ |
| $N_5$ | $V_{(5,s)}$ | $]-\frac{4p}{5},\frac{p}{5}[$ | $(1,\frac{p}{5}-s)$ | $(11,\frac{p}{5}-s)$ |
| $\bar{N}_5$ | $V_{(5,s)}$ | $]-\frac{p}{5},\frac{4p}{5}[$ | $(11,\frac{p}{5}+s)$ | $(1,\frac{p}{5}+s)$ |
| $N'_5$ | $V_{(5,s)}$ | $]-\frac{3p}{5},\frac{2p}{5}[$ | $(7,\frac{2p}{5}-s)$ | $(17,\frac{2p}{5}-s)$ |
| $\bar{N}'_5$ | $V_{(5,s)}$ | $]-\frac{2p}{5},\frac{3p}{5}[$ | $(17,\frac{2p}{5}+s)$ | $(7,\frac{2p}{5}+s)$ |

$$(3.9)$$

The simple current $N_{14}$ generates a $\mathbb{Z}_2$ symmetry whose orbits all have length 2:

$$\{D_1, N_{14}\}, \{D_7, N_8\}, \{D_{11}, N_4\}, \{D_{13}, N_2\}, \{N_3, N'_5\}, \{N_9, N_5\} . \tag{3.10}$$

Modulo this simple current symmetry, and modulo, parity, the fusion rules are

$$D_7 * D_7 = D_1 + D_7 + D_{11} + D_{13} + N_3 + \bar{N}_3 + N_5 + \bar{N}_5 \ , \tag{3.11a}$$

$$D_7 * D_{11} = {}^e D_{13} + {}^e N_2 + N_3 + \bar{N}_3 + N_5' + \bar{N}_5' \ , \tag{3.11b}$$

$$D_7 * D_{13} = D_7 + {}^e D_{11} + {}^e D_{13} + {}^e N_2 + N_3 + \bar{N}_3 + {}^e N_4 + N_5' + \bar{N}_5' \ , \tag{3.11c}$$

$$D_7 * N_3 = D_7 + D_{11} + D_{13} + N_2 + N_3 + \bar{N}_3 + N_4 + N_5 + N_5' + N_9 \ , \tag{3.11d}$$

$$D_7 * N_9 = N_3 + N_5' + N_8 \ , \tag{3.11e}$$

$$D_{11} * D_{11} = D_1 + {}^e N_4 + \bar{N}_5 + N_5 + \bar{N}_9 + N_9 \ , \tag{3.11f}$$

$$D_{11} * D_{13} = {}^e D_7 + {}^e D_{13} + {}^e N_2 + {}^e \bar{N}_3 + {}^e N_3 + {}^e \bar{N}_5' + {}^e N_5' + N_8 \ , \tag{3.11g}$$

$$D_{11} * N_3 = D_7 + {}^e D_{13} + {}^e N_2 + {}^e \bar{N}_3 + N_3 + {}^e \bar{N}_5' + N_5' + N_8 \ , \tag{3.11h}$$

$$D_{11} * N_9 = D_{11} + N_4 + \bar{N}_9 \ , \tag{3.11i}$$

$$D_{13} * N_3 = D_7 + {}^e D_{11} + D_{13} + {}^E N_2 + {}^e \bar{N}_3 + N_3 + {}^e N_4 + {}^e \bar{N}_5' + \bar{N}_5 + N_8 + \bar{N}_9 \ , \tag{3.11j}$$

$$D_{13} * N_9 = D_{13} + N_2 + \bar{N}_3 + \bar{N}_5' \ , \tag{3.11k}$$

$$N_3 * N_9 = D_7 + \bar{N}_3 + N_5' + N_8 \ , \tag{3.11l}$$

$$N_3 * N_3 = D_1 + D_7 + D_{11} + D_{13} + N_3 + \bar{N}_3 + N_4 + N_5 + \bar{N}_5 + \bar{N}_5' + N_8 + N_9 \ , \tag{3.11m}$$

$$N_9 * N_9 = D_1 + N_5 + N_9 \ , \tag{3.11n}$$

$$D_{13} * D_{13} = D_1 + {}^e D_7 + {}^e D_{11} + D_{13} + N_3 + \bar{N}_3 + {}^e N_4$$
$$+ N_5 + \bar{N}_5 + {}^E N_5' + {}^e \bar{N}_5' + N_8 + N_9 + \bar{N}_9 \ , \tag{3.11o}$$

$$N_3 * \bar{N}_3 = D_7 + {}^e D_{11} + {}^e D_{13} + {}^e N_2 + N_3 + \bar{N}_3 + {}^e N_4 + N_5' + \bar{N}_5' \ , \tag{3.11p}$$

where we have 2 instance of the notation $^E$ which indicates that only even couplings appear, whereas the more common notation $^e$ means that both even and odd couplings appear.

In particular, some terms are absent due to the extended symmetry (2.20), namely

$$C_{11,11,2} = C_{11,11,13} = C_{11,11,3} = C_{11,11,5'} = C_{11,11,7} = C_{11,11,8} = 0 \ . \tag{3.12}$$

To get the full consequences of the extended symmetry, we should act on these couplings as in Eq. (1.15), using the simple current $\mathbb{Z}_2$ symmetry (3.10).

Permutation invariance implies $C_{X,X,2} = 0$. Then, by $\mathbb{Z}_2$ symmetry, $C_{3,3,2} = 0 \implies C_{13,3,5'} = 0$. Furthermore, using the relation between integer and fractional Kac indices in Table (3.9), we can translate the permutation invariance condition (2.21) into a condition on odd or even couplings, and we find the constraints

$$C_{X,X,9}^{\text{even}} = C_{X,X,5}^{\text{even}} = C_{X,X,3}^{\text{even}} = C_{X,X,5'}^{\text{odd}} = 0 \ . \tag{3.13}$$

The first two constraints already follow from null vectors, but the last two lead to the nontrivial vanishings

$$C_{13,13,3}^{\text{even}} = C_{13,2,5'}^{\text{even}} = C_{2,2,3}^{\text{even}} = C_{13,13,5'}^{\text{odd}} = C_{13,2,3}^{\text{odd}} = C_{2,2,5'}^{\text{odd}} = 0 \ . \tag{3.14}$$

There remains one coupling whose vanishing is a purely numerical finding, without any motivation from the constraints we have discussed:

$$\boxed{C_{13,13,13}^{\text{even}} = 0} \ . \tag{3.15}$$

This is formally similar to the vanishing (2.18) for the $E_{p,12}$ models, which also involves the diagonal field with the largest first Kac index. However, in the $E_{p,12}$ case, the

vanishing was explained by the extended symmetry. As this is our only unexplained vanishing, we have checked that it occurs in several 4-point functions. In the case $p = 7$, we have bootstrapped $\left\langle V_{\langle 13,1\rangle}^d V_{\langle 13,1\rangle}^d V_{\langle 13,1\rangle}^d V_{\langle 13,1\rangle}^d \right\rangle$, $\left\langle V_{\langle 11,1\rangle}^d V_{\langle 13,1\rangle}^d V_{\langle 13,1\rangle}^d V_{\langle 13,1\rangle}^d \right\rangle$ and $\left\langle V_{\langle 13,1\rangle}^d V_{\langle 13,1\rangle}^d V_{\langle 11,1\rangle}^d V_{\langle 3,\frac{4}{3}\rangle}^d \right\rangle$. As we increase the numerical precision, we find that the 4-point structure constant $D_{(13,1)}^{(s)}$ (which includes a factor $C_{13,13,13}^{\mathrm{odd}}$) converges to a finite value, while $D_{(13,6)}^{(s)}$ (which includes a factor $C_{13,13,13}^{\mathrm{even}}$) tends to zero. We have observed these tendencies up to about 38 significant digits.

# 4 Structure constants

## 4.1 General form

Admitting that numerical bootstrap calculations give us numerical data for 4-point structure constants (2.3), the challenge is to extract analytic formulas for 3-point structure constants. To do this, we will first divide the numerical results by a reference structure constant, the same as in loop models [8]. After that, again as in loop models, we will asssume that we obtain simple functions of

$$ n = -2\cos\left(\pi\frac{p}{q}\right) . \tag{4.1} $$

This assumption is well-motivated in loop models, where $n$ is interpreted as the statistical loop weight. We do not have such a motivation in E-series minimal model, although their lattice description [3] might help if it could be generalized to all values of $p$. Nevertheless, the assumption will turn out to be obeyed by all our numerical results.

Let us first write the reference 3-point structure constant:

$$ C_{(r_1,s_1)(r_2,s_2)(r_3,s_3)}^{\mathrm{ref}} = \prod_{\epsilon_1,\epsilon_2,\epsilon_3=\pm} \Gamma_\beta^{-1}\left(\tfrac{\beta+\beta^{-1}}{2} + \tfrac{\beta}{2}\left|\sum_i \epsilon_i r_i\right| + \tfrac{\beta^{-1}}{2}\sum_i \epsilon_i s_i\right) , \tag{4.2} $$

where $\beta = \sqrt{\frac{p}{q}}$, and the term $|\sum_i \epsilon_i r_i|$ involves an absolute value. This ansatz is valid for 3-point functions of non-diagonal fields in the notation $V_{(r,s)}$ (1.7). It is also valid if diagonal fields are present, provided we write a diagonal field with left and right dimensions $\Delta = \bar{\Delta} = \frac{c-1}{24} + P^2$ using the non-diagonal notation $V_{(0,2\beta P)}$. We then deduce reference 2-point and 4-point structure constants via Eqs. (2.2) and (2.3). After dividing structure constants by their reference versions, we obtain the **reduced structure constants**

$$ b_i = \frac{B_i}{B_i^{\mathrm{ref}}} \quad , \quad c_{i,j,k} = \frac{C_{i,j,k}}{C_{i,j,k}^{\mathrm{ref}}} \quad , \quad d_k^{(s)} = \frac{D_k^{(s)}}{D_k^{(s),\mathrm{ref}}} . \tag{4.3} $$

By construction, reference structure constants obey the constraints from interchiral symmetry modulo signs. As a result,

$$ \frac{c_{(r_1,s_1+1)(r_2,s_2)(r_3,s_3)}}{c_{(r_1,s_1-1)(r_2,s_2)(r_3,s_3)}} = (-1)^{\max(2r_1,2r_2,2r_3,r_1+r_2+r_3)} , \tag{4.4} $$

$$ \frac{c_{(r_1,s_1+1)(r_2,s_2)(r_3,s_3)}}{c_{(r_1,s_1)(r_2,s_2)(r_3,s_3+1)}} = \begin{cases} (-1)^{r_1+r_2} & \text{if } r_2 \geq |r_1 - r_3| , \\ (-1)^{r_3} & \text{else} . \end{cases} \tag{4.5} $$

Let us now discuss the dependence on $n$ (4.1). For $p \in \mathbb{N}$ such that $p$ and $q$ are coprime, this number takes finitely many values, which are the roots of an even polynomial $P_q$ with

integer coefficients:

$$
\begin{array}{c|l}
q & P_q(n) \\
\hline
12 & n^4 - 4n^2 + 1 \\
18 & n^6 - 6n^4 + 9n^2 - 3 \\
30 & n^8 - 7n^6 + 14n^4 - 8n^2 + 1
\end{array}
\tag{4.6}
$$

Two of these polynomials have simple expressions in terms of Chebyshev polynomials of the first kind, namely $P_{12}(n) = 2T_4(\frac{n}{2}) - 1$ and $P_{18}(n) = 2T_6(\frac{n}{2}) - 1$. Any rational function of $n$ may be rewritten as a polynomial of $n$ of degree $\deg P_q - 1$. The analogy with loop models leads us to the conjecture:

> **Conjecture:** Reduced structure constants are polynomial functions of $n$ with values in the field $\mathbb{Q}(\cos(\frac{\pi}{q}))$.

This does not mean that reduced structure constants are polynomial with rational coefficients, i.e. elements of $\mathbb{Q}[n]$. For example, in the case $q = 12$, we have $n(n^2 - 3) = \pm\sqrt{2}$, with a $p$-dependent sign. The constant $\sqrt{2}$ is a polynomial function of $n$ with an irrational coefficient, and for all allowed $n$ its value belongs to the field $\mathbb{Q}(n) = \mathbb{Q}(\cos(\frac{\pi}{12}))$, but it cannot be rewritten as a polynomial with rational coefficients.

## 4.2 Explicit expressions for $q = 12$

Let us write the 4 values of $n$ in this case. We also introduce signs $\mu, \nu \in \{-1, 1\}$, which will be convenient for writing structure constants:

$$
\begin{array}{c||c|c|c|c}
p \bmod 24 & 1, 23 & 5, 19 & 7, 17 & 11, 13 \\
\hline
n & -\frac{\sqrt{3}+1}{\sqrt{2}} & -\frac{\sqrt{3}-1}{\sqrt{2}} & \frac{\sqrt{3}-1}{\sqrt{2}} & \frac{\sqrt{3}+1}{\sqrt{2}} \\
\hline
n^2 & 2+\sqrt{3} & 2-\sqrt{3} & 2-\sqrt{3} & 2+\sqrt{3} \\
\hline
\mu & 1 & -1 & 1 & -1 \\
\hline
\nu & 1 & 1 & -1 & -1
\end{array}
\tag{4.7}
$$

Equivalently, we have

$$
n = -\frac{\mu + \sqrt{3}\nu}{\sqrt{2}} \quad , \quad n^2 = 2 + \sqrt{3}\mu\nu \quad , \quad n^{-1} = \frac{\mu - \sqrt{3}\nu}{\sqrt{2}} \; .
\tag{4.8}
$$

Let us now write our results for 2-point and 3-point structure constants, where:

- We normalize fields such that $b_1 = b_4 = b_7 = 1$, but we find it convenient to have nontrivial $b_2$ and $b_3 = b_{\bar{3}}$. We include these 2-point structure constants in the following table under the names $c_{1,2,2}$ and $c_{1,3,3}$.

- For any 3-point structure constant $c_{r_1,r_2,r_3}$ we call $s_1, s_2, s_3$ the second Kac indices, whose values are displayed in Table (3.1).

- We display the sign that is picked under odd permutations of the 3 fields, which is $-1$ if the sum of conformal spins is odd.

- We display the condition on the parity of $\sum s_i = s_1 + s_2 + s_3$, under which each structure constant is valid. If all 3 fields are diagonal, this is simply $\sum s_i \in 2\mathbb{Z} + 1$ in the odd case and $\sum s_i + 1 \in 2\mathbb{Z} + 1$ in the even case. If non-diagonal fields are involved, this can be more complicated due to the map between integer and fractional Kac indices, see Table (3.1). We write superscripts $^{\text{even}}$ in even cases, and no superscript in odd cases.

- To obtain compact formulas, we find it convenient to write structure constants as powers of $n$ up to signs and constant prefactors, although in view of the polynomial equation $P_{12}(n) = 0$ there are many equivalent expressions.

| Name | Sign | $\in 2\mathbb{Z}+1$ | Value |
|---|---|---|---|
| $c_{1,2,2}$ | $1$ | $\sum s_i + 1$ | $-1$ |
| $c_{1,3,3}$ | $(-1)^{3s_2+3s_3}$ | $\sum s_i + \frac{2p}{3}$ | $-(-1)^{3s_3}\nu\sqrt{6}$ |
| $c_{2,4,7}$ | $-1$ | $\sum s_i - \frac{p}{2}$ | $-\sqrt{\frac{3}{2}} \cdot n^{-1}$ |
| $c_{2,4,7}^{\text{even}}$ | $-1$ | $\sum s_i + \frac{p}{2}$ | $\sqrt{\frac{3}{2}} \cdot n^{-1}$ |
| $c_{4,4,7}$ | $1$ | $\sum s_i$ | $\nu\sqrt{\frac{1}{2}}$ |
| $c_{4,4,7}^{\text{even}}$ | $1$ | $\sum s_i + 1$ | $\mu\sqrt{\frac{3}{2}}$ |
| $c_{2,2,7}$ | $1$ | $\sum s_i$ | $-\mu\nu\frac{1}{2} \cdot n^{-3}$ |
| $c_{2,2,7}^{\text{even}}$ | $1$ | $\sum s_i + 1$ | $-\sqrt{\frac{3}{4}} \cdot n^{-1}$ |
| $c_{7,7,7}$ | $1$ | $\sum s_i$ | $\nu\sqrt{2}$ |
| $c_{4,4,3}$ | $(-1)^{3s_3}$ | $\sum s_i - \frac{p}{3}$ | $(-1)^{s_1}\sqrt{2} \cdot n^{-1}$ |
| $c_{7,7,3}$ | $(-1)^{3s_3}$ | $\sum s_i - \frac{p}{3}$ | $(-1)^{s_1}\nu\sqrt{12} \cdot n^{-1}$ |
| $c_{7,3,\bar{3}}$ | $(-1)^{3s_2+3s_3}$ | $\sum s_i$ | $(-1)^{3s_3}\mu\sqrt{6} \cdot n^{-1}$ |
| $c_{3,3,3}$ | $1$ | $\sum s_i + 1$ | $(-1)^{\frac{1}{2}(s_1+s_2+s_3)}\mu\nu\sqrt{6} \cdot n^{-2}$ |
| $c_{2,2,3}$ | $(-1)^{3s_3}$ | $\sum s_i + \frac{2p}{3}$ | $(-1)^{\frac{1}{2}(s_1-s_2-3s_3)}\sqrt{\frac{1}{2}} \cdot n^{-3}$ |
| $c_{2,4,3}$ | $-(-1)^{3s_3}$ | $\sum s_i - \frac{5p}{6}$ | $-(-1)^{s_1+\frac{p}{2}}\mu\nu\sqrt{3} \cdot n^{-1}$ |

$$(4.9)$$

Apart from the 2-point structure constants $c_{1,2,2}$ and $c_{1,3,3}$, these are the 13 nonzero independent structure constants of the $E_{p,12}$ minimal models. After extracting these structure constants, we have tested them by directly checking crossing symmetry of a variety of 4-point functions [9].

## 4.3    Tests of the conjecture for $q = 18, 30$

The large spectrums of the $E_{p,18}$ and $E_{p,30}$ models lead to two difficulties when numerically computing structure constants:

1. There are many structure constants to be determined.

2. The decompositions of a given 4-point function into conformal blocks can involve many terms.

We will address these difficulties by testing the conjecture in a few 4-point functions only, and choosing 4-point functions that involve not too many different structure constants. These include 4-point functions where at least one of the 4 fields is a simple current. These also include 4-point functions where some or all 4 fields are identical, so that some or all of the 3 channels $s, t, u$ involve the same structure constants.

We numerically determine 4-point structure constants using the same methods as in loop models [2]. We then find analytic formulas under the assumption that the conjecture holds, i.e. that reduced structure constants are polynomial functions of $n$ (4.1). To find these polynomials, we compute the structure constants for a large enough number of values of $n$, equivalently of $p$. In practice we find it convenient to write our results as rational functions of $n$, rather than polynomials. Once we have an analytic formula, we compute the error, defined as

$$\text{Error} = \left| 1 - \frac{\text{numerical value}}{\text{analytic formula}} \right| . \tag{4.10}$$

We find that this error is always of the same order of magnitude as the deviation, which estimates the numerical uncertainty of our bootstrap method [2]. In particular, the error decreases when we push the numerics to better precision. In the results that we display, the error is computed in the case $p = 17$.

We will now display the comparison between the analytic structure constants and numerical results from the bootstrap, in the cases of structure constants with smallest second indices.

$\left\langle V_{\langle 5,1\rangle}^d V_{\langle 5,1\rangle}^d V_{\langle 5,1\rangle}^d V_{\langle 5,1\rangle}^d \right\rangle$ **for** $q = 18$

| Structure constant | Value | Error |
|:---:|:---:|:---:|
| $d_{\langle 1,1\rangle^d}^{(s,t,u)}$ | $1$ | $0$ |
| $d_{\langle 5,1\rangle^d}^{(s,t,u)}$ | $1$ | $10^{-40}$ |
| $d_{\langle 7,1\rangle^d}^{(s,t,u)}$ | $1$ | $10^{-40}$ |
| $d_{\langle 9,1\rangle^d}^{(s,t,u)}$ | $\frac{1}{2}$ | $10^{-37}$ |
| $d_{(3,\frac{1}{3})}^{(s,t,u)}$ | $\frac{6}{n^3}$ | $10^{-16}$ |

(4.11)

$\left\langle V_{\langle 5,1\rangle}^d V_{(8,\frac{1}{2})} V_{\langle 5,1\rangle}^d V_{(8,\frac{1}{2})} \right\rangle$ **for** $q = 18$

| Structure constant | Value | Error |
|:---:|:---:|:---:|
| $d_{\langle 1,1\rangle^d}^{(u)}$ | $1$ | $0$ |
| $d_{(4,\frac{1}{2})}^{(s,t)}$ | $18 \cdot \frac{8-3n^2}{5-11n^2}$ | $10^{-29}$ |

(4.12)

$\left\langle V_{(3,\frac{2}{3})} V_{(3,\frac{2}{3})} V_{(3,\frac{2}{3})} V_{(3,\frac{2}{3})} \right\rangle$ **for** $q = 18$

| Structure constant | Value | Error |
|---|---|---|
| $d^{(s,t,u)}_{\langle 1,s \rangle^d}$ | $1$ | $0$ |
| $d^{(s,t,u)}_{(3,\frac{2}{3})}$ | $-2n(n^2-4)$ | $10^{-37}$ |
| $d^{(s,t,u)}_{(4,-\frac{1}{2})}$ | $3 \cdot \frac{n^2-3}{n^2-1}$ | $10^{-21}$ |
| $d^{(s,t,u)}_{\langle 5,1 \rangle^d}$ | $-n^2 + 4$ | $10^{-39}$ |
| $d^{(s,t,u)}_{(8,-\frac{1}{2})}$ | $9 \cdot \frac{2n^2-9}{4-3n^2}$ | $10^{-13}$ |

$\qquad (4.13)$

$\left\langle V^d_{\langle 7,1 \rangle} V^d_{\langle 7,1 \rangle} V^d_{\langle 7,1 \rangle} V^d_{\langle 7,1 \rangle} \right\rangle$ **for** $q = 30$

| Structure constant | Value | Error |
|---|---|---|
| $d^{(s,t,u)}_{\langle 1,1 \rangle^d}$ | $1$ | $0$ |
| $d^{(s,t,u)}_{(3,\frac{14}{3})}$ | $-10 \cdot \frac{2n^4-10n^2+7}{9n^3}$ | $10^{-20}$ |
| $d^{(s,t,u)}_{(5,\frac{12}{5})}$ | $-62n^7 + 428n^5 - 828n^3 + 416n$ | $10^{-17}$ |
| $d^{(s,t,u)}_{\langle 7,1 \rangle^d}$ | $\frac{5}{3}$ | $10^{-21}$ |
| $d^{(s,t,u)}_{\langle 11,1 \rangle^d}$ | $1$ | $10^{-18}$ |
| $d^{(s,t,u)}_{\langle 13,1 \rangle^d}$ | $\frac{1}{3}$ | $10^{-15}$ |

$\qquad (4.14)$

$\left\langle V^d_{\langle 7,1 \rangle} V_{(14,\frac{1}{2})} V^d_{\langle 7,1 \rangle} V_{(14,\frac{1}{2})} \right\rangle$ **for** $q = 30$

| Structure constant | Value | Error |
|---|---|---|
| $d^{(u)}_{\langle 1,1 \rangle^d}$ | $1$ | $0$ |
| $d^{(s,t)}_{(8,\frac{1}{2})}$ | $-30\big(57n^6 - 389n^4 + 730n^2 - 329\big)$ | $10^{-48}$ |

$\qquad (4.15)$

# Acknowledgements

We are grateful to Ingo Runkel, Vincent Pasquier, Valentin Benedetti, and Yoshiki Fukusumi for enlightening discussions and correspondence. Moreover, we wish to thank Vincent Pasquier for helpful suggestions on the draft article.

This work is partly a result of the project ReNewQuantum, which received funding from the European Research Council.

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

*Conformal field theory* (8, 10)