# Peer review of "Fusion rules and structure constants of E-series minimal models"

_SciPost Physics, doi:SciPost Phys. 18, 163 (2025)_

## Round 2 · Referee Report · Connor Behan (Referee 1) · 2025-4-16

Report

A claim one often hears repeated is that Virasoro minimal models, the simplest conformal field theories in two dimensions, have been solved since the 1980s. As the present paper states in no uncertain terms, this is actually only true for the A-series and D-series. I.e. the Virasoro minimal models which consist only of diagonal primary operators as well as their $\mathbb{Z}_2$ orbifolds. The classic results for structure constants do not extend to the other modular invariant series known as the E-series. At the same time, a semi-analytic toolkit developed in recent years has led to exact solutions for a number of other critical models. These happen to have infinitely many primary operators and no unitarity which is an unreasonable thing to impose in statistical physics.

The authors of this paper address both of these efforts by using modern methods to compute structure constants in E-series minimal models thereby filling a gap in the literature. In this study of minimal models, which are labelled by relatively prime $p$ and $q$ for $q \in { 12, 18, 30 }$, results are most complete for $q = 12$. In this case, all structure constants are determined in closed form. To within tiny numerical errors, they are given by a double gamma functions times an element of $\mathbb{Q}(\cos(\pi/q))$. It therefore seems reasonable that this basic form continues to hold for $q = 18,30$ and the authors check this conjecture in a number of examples. Their original motivation, identifying a much larger set of (generically irrational and logarithmic) CFTs which contain these minimal models as special cases, is left as a problem for the future.

The final results of this paper were obtained with the help of a publicly accessible code. Its main purpose is to carry out the numerical bootstrap for a four-point function expressed in terms of Virasoro blocks where the spectrum has been specified as an input. This otherwise daunting task is sped up thanks to non-chiral fusion rules which can also be given as input. I.e. the set of OPE coefficients which may be shown to vanish analytically. For non-diagonal theories, these contain information which is much richer than what appears in the chiral fusion rules alone. Apart from constraints based on parity and single-valuedness which might be expected, simple currents, extended symmetries and interchiral symmetry also play a role in this analysis. The order of presentation is quite clear since section 2 discusses the various constraints, section 3 assembles the non-chiral fusion rules which follow from them, and section 4 covers the numerical results.

This is a careful and well motivated paper. It solves an open problem and further streamlines some CFT techniques which have been shown to have a wide range of applications. The following questions are mostly to check my understanding but I would encourage the authors to add remarks to the text about some of them before publication.

  1. The statement on p3 that A and D structure constants are known explicitly is given without a reference. The papers by Dotsenko and Fateev are often cited in other discussions of the A-series but it might be harder to find papers where the solution of the D-series can be found.

  2. Similarly, assuming it is not a historical accident, it would be helpful to know the main obstructions that prevented an E-series solution from being found earlier. In particular, the crossing matrix elements found by Dotsenko and Fateev are chiral results about the Virasoro algebra so it does not seem unreasonable to adapt their method of imposing trivial monodromy. I could imagine this being quite difficult if only the chiral fusion rules are known but maybe it is fruitful to use non-chiral fusion rules there as well.

  3. After p4 discusses extended symmetry for $q = 12$ and $q = 30$, it might be nice to comment on how one should interpret $\mathcal{R}{1, 1} \otimes \bar{\mathcal{R}}$ which is present for $q = 18$.

  4. Under eq (1.10), it would be best to either define $\beta_0^2$ or simply say that $p/q$ tends toward a value fixed by the central charge of the $PSU(n)$ model.

  5. I also found some wording in this paragraph to be too strong. Since the first two properties (gap above $r = 1$ going to infinity and the rest of the spectrm approaching $\mathcal{P}_{PSU(n)}$) are not conclusions of any argument, it does not make sense to say that you "therefore" expect a third property --- that the limit of $\tilde{E}_{p,q}$ is the $PSU(n)$ CFT. It is reasonable to expect that these properties are either all present or all absent. But they are quite a bit stronger than the conjecture that there is some $\tilde{E}_{p,q}$ such that the denominator of the second index increases with $q$.

  6. On p9 when you choose $r \leq q/2$ except when $q = 12$, is this because 12 is the only value of $q$ which is evenly even? If not, this sounds like a change of convention that would require explanation.

  7. Under eq (2.11), it looks strange to refer to the parity of $s_1 + s_2 + s_3$ when discussing non-diagonal fields whose $V_{(r, s)}$ notation allows $s$ to be fractional. Later sections seem to indicate that each term of $s_1 + s_2 + s_3$ here really refers to the value of $s$ one gets after converting the compact notation back to $V_{(r, s)(\bar{r}, s)}$. Stating this under eq (2.11) could be helpful.

  8. Section 2.1 says that any allowed even coupling also comes with odd couplings. And yet, section 2.2 designates a superscript E for cases taht only have an even coupling. It makes sense that the associated even coupling suggested by chiral fusion rules might later be seen to vanish a result of interchiral symmetry but this is a potential source of confusion.

  9. Looking at the last table in section 4.2, the reduced structure constants depend on the $s_i$ at most through the overall signs. Naively, this looks remarkable to me. If that is idneed the case, it would be fair to advertise this result more.

  10. Finally, some phrases in need of grammatical correction are "one a case-by-case basis", "have more often be written", "have 2 instance of" and "motivation in E-series minimal model".

Requested changes

See report.

Recommendation

Ask for minor revision

  • validity: -
  • significance: -
  • originality: -
  • clarity: -
  • formatting: -
  • grammar: -

Author:  Rongvoram Nivesvivat  on 2025-05-08  [id 5467]

(in reply to Report 1 by Connor Behan on 2025-04-16)

2.1. We have added the reference for the three-point functions in the A- and D-series in the second point of the 4 items on page, citing as "reference [2] and references therein".

2.2. On page 3, we have itemized why the E-series minimal models are not fully solved, and we have added reference [3], which discuss how to compute the models' three-point functions on a case-by-case basis.

2.3. At the end of Section 2.3 we have stated more explicitly that a simple current can be interpreted as an extended $\mathbb{Z}_2$ symmetry, and that Section 2.4 deals with extended symmetries that are not simple currents.

2.4. We have clarified this point.

2.5. We have added equations (1.11), (1.12), and (1.13) to clarified our speculation on obtaining the $PSU(n)$ CFT as non-rational limits.

2.6. On page 9, under the equation (2.7), we have clarified why we choose $r \leq q/2$ except for the case $q = 12$.

We make an exception for the case $q = 12$ because we want all three-point functions to obey our convention for the parity constraints for any $q$: any allowed even coupling also comes with an allowed odd coupling.

For $q = 12$, the case $r = 7$ belongs to the identity sector of the extended symmetry (1.6) whereas $r=5$ belongs to the $\epsilon$. This makes three-point functions for the case $q =12$ agree with our convention for the parity constraints.

2.7. We have clarified that the convention for the parity $s1 + s2 + s3$ under the equation (2.11). For this parity, we use the notations (1.5) wherein the Kac indices $s_i$ are integers for both diagonal and non-diagonal fields.

2.8. We have stressed that we write the superscript $e$ in the non-chiral fusion rules when both even and odd parities are allowed.

2.9. We have explained after (4.3) why the structure constants on the table (4.9) only depends on the second indices through signs.

2.10. We have corrected these typos.

Connor Behan  on 2025-05-08  [id 5469]

(in reply to Rongvoram Nivesvivat on 2025-05-08 [id 5467])

Thank you very much for the changes. My concerns have all been addressed now.

---

## Round 2 · Referee Report · Jiaxin Qiao (Referee 2) · 2025-4-21

Report

This paper studies the E-series minimal models within the ADE classification of two-dimensional rational conformal field theories. The spectrum of these models is known, but their OPE coefficients remain unsolved, unlike in the A- and D-series where both the spectrum and OPE coefficients are explicitly known. The authors use semi-analytic bootstrap techniques—a combination of analytical and numerical methods adapted from their previous work on critical loop models—to compute OPE coefficients. They fully solve the $q=12$ case (the $E_6$ series) and provide partial analytical/numerical results for $q=18$ (the $E_7$ series) and $q=30$ (the $E_8$ series).

The main results of this work include: 1. A derivation of non-chiral fusion rules for E-series minimal models, using constraints from null vectors, interchiral symmetry, simple currents, extended symmetries, permutation symmetry, and parity. This extends previous results by Rida and Sami on D-series minimal models.

  1. A conjecture that structure constants are given by a universal function involving double Gamma function, times polynomial functions of $n \equiv -2\cos(\pi p/q)$ with values in the field $\mathbb{Q}\left(\cos(\pi/q)\right)$. The conjecture is tested with high-precision numerics as described below.

  2. Exact analytic formulas for all OPE coefficients in the case of $q=12$. The formulas are guessed from numerical bootstrap results, consistent with the conjecture and confirmed with high-precision numerics.

  3. For $q=18$ and $q=30$, exact analytic formulas for a few four-point function structure constants are found from numerical results, and are consistent with the conjecture and high-precision numerics.

The work is a valuable and original contribution to the study of 2D conformal field theory, filling an important gap in the literature. Their combination of numerical and analytic methods is well-motivated and produces compelling results. The numerical-analytic agreement is impressive and thorough. The single unexplained vanishing structure constant in the $q=30$ case is carefully identified. I recommend this paper for publication in SciPost Physics.

I have the following comments and questions: 1. Is there an argument for the uniqueness of the OPE coefficients in the E-series minimal models? By uniqueness I mean that the structure constants are determined up to an overall sign convention: $C_{ijk} \rightarrow (-1)^{n_i + n_j + n_k} C_{ijk}$ corresponding to field redefinitions $V_i \rightarrow (-1)^{n_i} V_i$ with $n_i \in {0,1}$.

  1. The last term in equation (1.5a) appears to have a typo: $s$ should be odd there, otherwise the expression does not match Table 10.3 of the yellow book (Di Francesco et al.).

  2. After equation (1.5c), the text says "the spectrums are block diagonal." It should be clarified under which symmetry or transformation the spectrum is block diagonal.

  3. In the discussion of the PSU($n$) CFT on page 5, it is claimed that taking the limit $p, q \rightarrow \infty$ with $p/q \rightarrow \beta_0^2$ leads to the gap above $r = 1$ for diagonal fields $V_{\langle r,s \rangle}^d$ tending to infinity. But from equation (1.3), we get

    $$ \Delta_{r,s} - \Delta_{1,s} = \frac{1}{4} \beta_0^2(r^2 - 1) - \frac{1}{2}(r - 1)s $$
    in the limit. This does not diverge for finite $\beta_0$ and $r$. Is it implicitly assumed that $r$ grows with $q$?

  4. Below equation (2.8), the authors assume: “the non-chiral fusion rules of $V^d_{\langle 1,2 \rangle}$ follow from the chiral fusion rules of $\mathcal{R}_{1,2}$.” Is this an assumption, or is it a logical consequence of the operator content and chiral fusion rules?

  5. Before knowing the closed-form expressions, is extended symmetry used as an input to the numerical bootstrap? The discussion in the last paragraph of page 7 is a bit confusing. The first point suggests the spectrum is the only input, but the second point seems to say that symmetries are also inputs. The authors should clarify precisely what the bootstrap inputs are.

  6. Relatedly, what happens if extended symmetry and other non-standard constraints in section 2 are not imposed? Do they merely explain vanishing structure constants a posteriori, or are they needed to uniquely determine a consistent solution to the bootstrap equations?

  7. At the end of section 4.2, the authors refer to $c_{1,2,2}$ and $c_{1,3,3}$ as “2-point structure constants.” According to equation (2.1), this is not quite appropriate terminology, since these coefficients also involve $V_{\langle1,s\rangle}$ with $s\neq1$. The phrasing could be clarified.

Recommendation

Publish (surpasses expectations and criteria for this Journal; among top 10%)

  • validity: top
  • significance: top
  • originality: top
  • clarity: top
  • formatting: excellent
  • grammar: excellent

Author:  Rongvoram Nivesvivat  on 2025-05-08  [id 5466]

(in reply to Report 2 by Jiaxin Qiao on 2025-04-21)

1.1. Our numerical bootstrap results imply that the E-series minimal models' spectra always yield a unique solution to the crossing-symmetry equation. Therefore, this allows us to uniquely factorize the 4-point structure constants into the 3-point structure constants up to some field renormalizations. (This uniqueness is a numerical result, we do not have a proof.)

We have added these details at the beginning of Section 4.1.

1.2. We have corrected the parity of $s$ in last term of (1.5a) as suggested.

1.3. The spectrum for the case $q =12$ is block-diagonal w.r.t to the Ising category whereas the spectrum for the case $q =30$ is block-diagonal w.r.t to the Lee-Yang category.

This has been clarified after equation (1.6).

1.4. To obtain the PSU(n) CFT, we need to assume that the first Kac indices $r$ increases as $q$ is increased.

To make our speculation clearer, we have added the equations (1.11), (1.12), and (1.13), which speculate how to obtain the $PSU(n)$ CFT as the non-rational limit of generalizations of E-series minimal models.

1.5. We have clarified that the non-chiral fusion rules of $V_{1,2}^d$ are not an assumption.

The non-chiral fusion rules of $V_{1,2}^d$ can be fully determined by using the constraints from the null vectors, the crossing-symmetry equation, and the single-valuedness. Then, the resulting non-chiral fusion rules of $V_{1,2}^d$ produce the primary fields that are allowed by the chiral fusion rules of $R_{1, 2}$.

1.6. For computing the non-chiral fusion rules, the bootstrap uses the spectrum and the constraints from the null vectors (chiral fusion rules) as the only inputs. Then, we check that the resulting non-chiral fusion rules agree with the extended symmetries and the non-standard constraints in Section 2. For deducing the analytic structure constants, the inputs to the bootstrap are the non-chiral fusion rules, which reduce the number of unknowns in the crossing-symmetry equation and allow us to easily access numerical results at high-precision. All of the above has been clarified at the end of Section 1.4, after equation (1.18).

1.7. See point 1.6.

1.8. We have rewritten 2-point structure constants as simply structure constants since $c_{1, 2, 2}$ and $c_{1, 3, 3}$ on the table (4.9) are the structure constants of the three-point functions $<V_{1,s}V_{2,s'}V_{2,s''} $> and $<V_{1,s}V_{3,s'}V_{3,s''} >$.

Jiaxin Qiao  on 2025-05-14  [id 5477]

(in reply to Rongvoram Nivesvivat on 2025-05-08 [id 5466])

Thanks, I am fine with the revised draft.

---

## Round 3 · List of Changes

Changes according to Jiaxin Qiao's report

  1. Our numerical bootstrap results imply that the E-series minimal models' spectra always yield a unique solution to the crossing-symmetry equation. Therefore, this allows us to uniquely factorize the 4-point structure constants into the 3-point structure constants up to some field renormalizations. (This uniqueness is a numerical result, we do not have a proof.)

We have added these details at the beginning of Section 4.1.

  1. We have corrected the parity of $s$ in last term of (1.5a) as suggested.

  2. The spectrum for the case $q =12$ is block-diagonal w.r.t to the Ising category whereas the spectrum for the case $q =30$ is block-diagonal w.r.t to the Lee-Yang category.

This has been clarified after equation (1.6).

  1. To obtain the PSU(n) CFT, we need to assume that the first Kac indices $r$ increases as $q$ is increased.

To make our speculation clearer, we have added the equations (1.11), (1.12), and (1.13), which speculate how to obtain the $PSU(n)$ CFT as the non-rational limit of generalizations of E-series minimal models.

  1. We have clarified that the non-chiral fusion rules of $V_{1,2}^d$ are not an assumption.

The non-chiral fusion rules of $V_{1,2}^d$ can be fully determined by using the constraints from the null vectors, the crossing-symmetry equation, and the single-valuedness. Then, the resulting non-chiral fusion rules of $V_{1,2}^d$ produce the primary fields that are allowed by the chiral fusion rules of $R_{1, 2}$.

  1. For computing the non-chiral fusion rules, the bootstrap uses the spectrum and the constraints from the null vectors (chiral fusion rules) as the only inputs. Then, we check that the resulting non-chiral fusion rules agree with the extended symmetries and the non-standard constraints in Section 2. For deducing the analytic structure constants, the inputs to the bootstrap are the non-chiral fusion rules, which reduce the number of unknowns in the crossing-symmetry equation and allow us to easily access numerical results at high-precision. All of the above has been clarified at the end of Section 1.4, after equation (1.18).

  2. See point 6.

  3. We have rewritten 2-point structure constants as simply structure constants since $c_{1, 2, 2}$ and $c_{1, 3, 3}$ on the table (4.9) are the structure constants of the three-point functions $<V_{1,s}V_{2,s'}V_{2,s''} $ and $<V_{1,s}V_{3,s'}V_{3,s''} $.

Changes according to Connor Behan's report

  1. We have added the reference for the three-point functions in the A- and D-series in the second point of the 4 items on page, citing as "reference [2] and references therein".

  2. On page 3, we have itemized why the E-series minimal models are not fully solved, and we have added reference [3], which discuss how to compute the models' three-point functions on a case-by-case basis.

  3. At the end of Section 2.3 we have stated more explicitly that a simple current can be interpreted as an extended $\mathbb{Z}_2$ symmetry, and that Section 2.4 deals with extended symmetries that are not simple currents.

  4. We have clarified this point.

  5. We have added equations (1.11), (1.12), and (1.13) to clarified our speculation on obtaining the $PSU(n)$ CFT as non-rational limits.

  6. On page 9, under the equation (2.7), we have clarified why we choose $r \leq q/2$ except for the case $q = 12$.

We make an exception for the case $q = 12$ because we want all three-point functions to obey our convention for the parity constraints for any $q$: any allowed even coupling also comes with an allowed odd coupling.

For $q = 12$, the case $r = 7$ belongs to the identity sector of the extended symmetry (1.6) whereas $r=5$ belongs to the $\epsilon$. This makes three-point functions for the case $q =12$ agree with our convention for the parity constraints.

  1. We have clarified that the convention for the parity $s1 + s2 + s3$ under the equation (2.11). For this parity, we use the notations (1.5) wherein the Kac indices $s_i$ are integers for both diagonal and non-diagonal fields.

  2. We have stressed that we write the superscript $e$ in the non-chiral fusion rules when both even and odd parities are allowed.

  3. We have explained after (4.3) why the structure constants on the table (4.9) only depends on the second indices through signs.

  4. We have corrected these typos.

---

## Editorial Decision

published